# The Inhibitory Effects of Plant Derivate Polyphenols on the Main Protease of SARS Coronavirus 2 and Their Structure–Activity Relationship

**DOI:** 10.3390/molecules26071924

**Published:** 2021-03-30

**Authors:** Thi Thanh Hanh Nguyen, Jong-Hyun Jung, Min-Kyu Kim, Sangyong Lim, Jae-Myoung Choi, Byoungsang Chung, Do-Won Kim, Doman Kim

**Affiliations:** 1Institutes of Green Bioscience and Technology, Seoul National University, Pyeongchang-gun 25354, Gang-won-do, Korea; hara2910@snu.ac.kr; 2Radiation Research Division, Korea Atomic Engery Research Institute, Jeongeup 56212, Jeollabuk-do, Korea; Jungjh83@kaeri.re.kr (J.-H.J.); mkkim@kaeri.re.kr (M.-K.K.); saylim@kaeri.re.kr (S.L.); 3Ottogi Sesame Mills Co., Ltd., Eumseong-gun 27623, Chungcheongbuk-do, Korea; mjchoi@ottogism.co.kr (J.-M.C.); bschung@ottogism.co.kr (B.C.); 4Department of Physics, Gangneung-Wonju National University, Gangneung 25457, Gangwon-do, Korea; Do.Won.Kim@cern.ch; 5Graduate School of International Agricultural Technology, Seoul National University, Pyeongchang-gun 25354, Gangwon-do, Korea

**Keywords:** main protease, SARS-CoV-2, polyphenol, kinetic, structure–activity relationship, black garlic

## Abstract

The main protease (M^pro^) is a major protease having an important role in viral replication of the severe acute respiratory syndrome coronavirus 2 (SARS-CoV-2), the novel coronavirus that caused the pandemic of 2020. Here, active M^pro^ was obtained as a 34.5 kDa protein by overexpression in *E. coli* BL21 (DE3). The optimal pH and temperature of M^pro^ were 7.5 and 37 °C, respectively. M^pro^ displayed a *K_m_* value of 16 μM with Dabcyl-KTSAVLQ↓SGFRKME-Edans. Black garlic extract and 49 polyphenols were studied for their inhibitory effects on purified M^pro^. The IC_50_ values were 137 μg/mL for black garlic extract and 9–197 μM for 15 polyphenols. The mixtures of tannic acid with puerarin, daidzein, and/or myricetin enhanced the inhibitory effects on M^pro^. The structure–activity relationship of these polyphenols revealed that the hydroxyl group in C3′, C4′, C5′ in the B-ring, C3 in the C-ring, C7 in A-ring, the double bond between C2 and C3 in the C-ring, and glycosylation at C8 in the A-ring contributed to inhibitory effects of flavonoids on M^pro^.

## 1. Introduction

A new coronavirus disease, termed coronavirus disease 2019 (COVID-19), was first identified in Wuhan, China in December 2019 (Hasan et al., 2020). COVID-19 has spread to 235 countries resulting in 81,947,503 confirmed cases and 1,808,041 confirmed deaths by 2 January 2020 according to the World Health Organization (WHO) (covid19.who.int) (accessed on 2 January 2021) and the infection and mortality is increasing day by day. Due to rapid dissemination and deaths, COVID-19 was declared a pandemic for the first time by WHO on 11 March 2020 [1]. COVID-19 has become a global public health crisis, devastating the global economy, and affecting all stages of the education system [2]. Imposing social distancing and self-isolation, restricting travel, closing school, wearing masks, improving immunity, and using air-disinfectant and surface-sanitizing agents are current methods to limit COVID-19 infection [2,3] due to its human-to-human transfer via aerosol transmission [4].

COVID-19 is caused by a novel severe acute respiratory syndrome-related coronavirus (SARS-CoV-2) [5] which belongs to the β-coronavirus type and is closely related to MERS-CoV and SARS-CoV [6,7]. Like other members of coronaviruses, the genome of SARS-CoV-2 is a positive sense, single-stranded RNA (~30 kb) enveloped and contains at least 6 open reading frames (ORFs) [7]. When the virus infects the cell, its genome acts as the mRNA and initiates the synthesis of polyproteins pp1a and pp1b which are essential for viral replication and transcription [8]. These polyproteins encode by two proteases called main protease (M^pro^) or 3C-like protease and papain-like protease [8]. Thus, M^pro^ is considered one of the important targets for blocking viral replication. At present, there are no antiviral drugs on the market for COVID-19 treatment [9]. Broad-spectrum antibiotics such as amoxicillin, azithromycin, and fluoroquinolones, antiviral drugs such as ritonavir, remdesivir, oseltamivir, chloroquine, and lopinavir, corticosteroids such as methylprednisolone, and convalescent plasma are combined to use for COVID-19 treatment [10,11]. More than 200 clinical trials are ongoing (clinicaltrials.gov) (accessed on 11 October 2020).

Nutraceuticals are foods that provide medical or health benefits such as avoiding side effects, protecting human health against chronic diseases, improving human health, and are economically affordable [12,13,14,15]. In addition, herbal medicines have been used for COVID-19 treatment in China [16]. Herbal medicines not only showed the prevention of COVID-19 infection in healthy persons but also improved the health state of treated patients [16]. While screening nutrients and herbal medicines for COVID-19, we found that black garlic showed inhibitory effects on COVID-19 M^pro^. Black garlic has a higher total phenolic and total flavonoid content than raw garlic [17]. Black garlic contains various phenolic acids including gallic acid, vanillic acid, chlorogenic acid, caffeic acid, ferulic acid, and coumaric acid), and flavonoids including catechin, epicatechin, epigallocatechin (EGCG), quercitrin, myricetin, resveratrol, morin, and quercetin [17]. Besides the antioxidant, cardioprotection, anti-cancer, anti-inflammatory, anti-aging, and protective activities, polyphenols have antiviral activities against enterovirus, influenza virus, SARS-CoV, Zika, and dengue virus [18,19,20,21]. In our previous study, polyphenols such as ampelopsin, puerarin, quercetin, daidzein, epigallocatechin gallate (EGCG), and gallocatechin gallate (GCG) were shown to inhibit the 3CL^pro^ of SARS-CoV [20]. M^pro^ sequence identity between SARS-CoV and SARS-CoV-2 was found to be 96% [22]. Also, the inhibitory effects of EGCG and theaflavin on M^pro^ of SARS-CoV-2 were confirmed [23]. Therefore, it is essential to study the inhibitory effects of each polyphenol on SARS-CoV-2 M^pro^ and their structure-activity relationship. In this study, the M^pro^ of SARS-CoV-2 was overexpressed in *E. coli* BL21(DE3) using IPTG as an inducer and purified using Ni-NTA chromatography. Its proteolytic activity was confirmed by using Dabcyl-KTSAVLQSGFRKME-Edans (FRET) as a substrate. The optimum pH, pH stability, optimum temperature, and temperature stability of SARS-CoV-2′s M^pro^ were studied. Also, the Michaelis–Menten constant (*K_m_*) and the effect of DMSO concentration on M^pro^ activity were studied. Then, the inhibitory effects of black garlic extract and 49 polyphenols on M^pro^ and their structure–activity relationships were investigated. The 50% inhibitory concentration (IC_50_) of black garlic and 15 polyphenols were determined. In addition, the effects of mixtures of polyphenols (tannic acid, puerarin, daidzein, and myricetin) on M^pro^ activity were studied.

## 2. Results

### 2.1. Biochemical Characterization of SARS-CoV-2 M^pro^

The nucleotide sequences encoding SARS-CoV-2 M^pro^ were optimized for expression in *E. coli* through a custom gene synthesis service of Genscript. It was expressed in *E. coli* BL21(DE3) by using 1.0 mM IPTG at 16 °C for 12 h, purified using Ni-Sepharose chromatography, and confirmed by SDS-PAGE (Figure 1A). A 34.5 kDa protein was observed on 12% SDS-PAGE (Figure 1A). It had activity with Dabcyl-KTSAVLQ↓SGFRKME-Edans, a SARS-CoV 3CL^pro^ substrate (Nguyen et al., 2012) in 50 mM Tris-HCl pH 7.5. The purified M^pro^ reached maximum activity at pH 7.5 (Figure 1B). M^pro^ showed over 90% activity at the pH from 5.6 to 9.9 after 24 h incubation at 4 °C. Maximum activity of M^pro^ was obtained at 37 °C (Figure 1C). A thermos-stability assay showed that M^pro^ was stable at 37 °C (Figure 1C), then it was left at 76% activity when kept at 45 °C for 24 h. M^pro^ activity was not detected when kept at 60 °C for 24 h (Figure 1C). The *K_m_* value of M^pro^ calculated from the Lineweaver–Burk double reciprocal plot was 16 μM with Dabcyl-KTSAVLQSGFRKME-Edans (Figure 1D and Appendix A).

### 2.2. Influence of Dimethylsulfoxide on M^pro^ Activity

Dimethylsulfoxide (DMSO) is a potent organic solvent that dissolves a variety of organic compounds due to its high dielectric constant and stereochemistry [24]. Although DMSO has been used as an additive, drug carrier to cells, and a cryoprotector, it can affect enzyme activity [24,25]. Therefore, the effect of DMSO on M^pro^ activity was studied (Figure 2). M^pro^ was stable in the presence of up to 10% (*v/v*) DMSO (Figure 2). Then, its activity was reduced when the DMSO concentration increased over 10% (*v/v*). At 50% (*v/v*) DMSO, M^pro^ activity was not detected (Figure 2).

### 2.3. Inhibitory Effects of Plant Derivative Polyphenols on SARS-CoV-2 M^pro^

Polyphenols are bioactive compounds found in fruits, vegetables, grains, and herbs and have been widely studied owing to their nutraceutical activity, such as anti-bacterial activities, anti-viral activities, anti-cancer, anti-inflammatory, and anti-diabetes [26]. Polyphenols have been divided into many classes depending on their strength ring. Lignans, phenolic alcohols, stiblins, phenolic acids, and flavonoids are the main polyphenol classes [27]. In this study, we found that black garlic extract inhibited 100% SARS-CoV2 M^pro^ activity at 0.5 mg/mL. The IC_50_ value of extracted garlic acid was 137 ± 10 μg/mL (Table 1). Black garlic is produced by heating raw garlic at high temperatures. Black garlic contains different phenolic acids such as gallic acid, caffeic acid, vanillic acid, ferulic acid, and chlorogenic acid, and various flavonoids such as epicatechin, catechin, epigallocatechin gallate, resveratrol, myricetin, and quercetin [17]. Thus, we examined the inhibitory effects of 49 polyphenols from different classes on M^pro^. The chemical structures of 49 polyphenols and their inhibitory activity on M^pro^ are shown in Figure 3 and Table 1. Among them, caffeine, capsaicin, teniposide, and idebenone did not inhibit M^pro^ at 200 μM. M^pro^ inhibitory effects of kaempferol, quercetin-4′-*O*-α-d-glucopyranoside, naringin, epicatechin, catechin, chrysin, trigonelline, ascorbic acid, hydroquinone, gallic acid, pyrogallol, and catechol were less than 20% compared to the control while the other compounds including astragalin, myricein, quercetin, quercetagenin, ampelopsin, ampelopsin-4′-*O*-α-d-glucopyranoside, naringenin, epigallocatechin gallate (EGCG), vitexin, daidzein, puerarin, resveratrol, tannic acid, chlorogenic acid, and caffeic acid showed over 50% inhibitory activity on M^pro^. The compounds exhibiting more than 50% inhibitory activity on M^pro^ were selected to determine IC_50_. Their IC_50_ values were from 9 to 197 μM (Table 1).

### 2.4. Structure-Activity Relationship of Plant Derivative Polyphenols against SARS-CoV-2 M^pro^

First, the inhibitory activity of compounds in the sample group was compared. In the flavonol group, the order of M^pro^ inhibition activity was as follows: kaempferol < quercetin-4′-*O*-α-d-glucopyranoside < rutin < quercetagenin < astragalin < quercetin < myricetin at 200 μM. Flavonols are a class of flavonoids that has a 3-hydroflavone backbone. The effects of hydroxyl group (OH) substitution in the B-ring on M^pro^ inhibitory effects were evaluated. The inhibitory effects of quercetin, which has 3′-OH and 4′-OH at the B-ring, and kaempferol, which has 4′-OH group at the B-ring, were lower than that of myricetin which has 3′-OH, 4′-OH, and 5′-OH groups at the B-ring. The absence of 3′-OH and 4′-OH groups at the B-ring was the reason for the lower inhibitory activity of kaempferol and quercetin than myricetin. When quercetin was glycosylated at C4′ in the B-ring (quercetin-4′-*O*-α-glucopyranoside) and C3 in the C-ring (rutin), its inhibitory effect was decreased. The presence of the OH group at C6 in the A-ring of quercetagenin decreased the inhibitory activity compared to quercetin. Therefore, the OH at C6 in the A-ring, the C3′, C4′, C5′ in the B-ring, and the C3 in C-ring affected M^pro^ inhibitory activity. Ampelopsin is dihyromyricetin. The existence of double bonds between C2 and C3 in the C-ring of myricetin was the reason for the higher inhibitory activity of myricetin than ampelopsin.

In the flavanone group, the order of M^pro^ inhibitory activity was as follows: naringin < hesperidin < naringenin. Naringenin is glycosylated naringin. However, its inhibitory activity was 3.2-fold higher than that of naringin. Hesperidin that contained glycosylation at 7-OH at the A-ring like the naringenin and the methoxy group at position 5′ of the B-ring was shown to have higher inhibitory activity than that of naringin but lower inhibitory activity than that of naringenin, indicating that glycosylation at the C7 position enhanced the M^pro^ inhibitory effect. In contrast, the methoxy group at C5′ in the B-ring reduced its inhibitory activity.

In the flavan-3-ols group, the order of the inhibitory effect was as follows: epicatechin (EC) < catchin < epicatechin gallate (ECG) < catechin gallate (CG) < epigallocatechin (EGC) < gallocatechin gallate (GCG) < epigallocatechin gallate (EGCG) (Table 1). EGCG, GCG, and ECG which contained three OH groups at C3′, C4′, and C5′ in the B-ring showed higher M^pro^ inhibitory activity than that of the remaining compounds (EC, catechin, CG, and EGC), suggesting that the OH group at C3′, C4′, C5′ in the B-ring increased the inhibitory activity against M^pro^, similar to the compounds in the flavonol group. EGCG, GCG, CG, and ECG, which have a galloyl moiety at C3 in the C-ring, showed higher M^pro^ inhibitory activity, suggesting that galloyl moiety at C3 of the C-ring increased the inhibitory activity on M^pro^. Thus, the inhibitory activity of gallic acid on M^pro^ was studied. Gallic acid inhibited 6.9% M^pro^ activity at 200 μM. Besides gallic acid, we also tested the inhibitory effects of tannic acid composed of 10 galloyl moieties. Tannic acid inhibited 100% M^pro^ activity and its IC_50_ value was 9 μM. Compared to myricetin, EGCG, which has the same three OH groups in the B-ring, galloy moiety at C3 in the C-ring, no double bonds between C2 and C3, and no C4 = O in the C-ring, displayed decreased inhibitory activity on M^pro^.

In the flavone group, the order of inhibitor activity on M^pro^ was as follows: chrysin < apigenin < luteolin < vitexin. Chrysin, lacking the OH group in the B-ring, exhibited the lowest inhibitory activity on M^pro^. Compared to the three compounds (luteolin, chrysin, and apigenin), we found that the more the OH group increased in the B-ring, the more the inhibitory effects on M^pro^ increased. While with attacked glucosyl moiety at C8 in the A-ring, the inhibitory activity on M^pro^ increased by 2-fold that of apigenin. The difference between flavone and isoflavone is the linked position of B-ring to C-ring. Compared to apigenin, which contained the same OH group in the B-ring but the B-ring was linked in position C3 in the C-ring and lacked the OH group at C5 in the A-ring, daidzein showed an increased inhibition effect on M^pro^. Puerarin, which contained the 8-C-glucoside of daidzein, exhibited a slightly higher inhibitory effect on M^pro^ than that of daidzein. From the inhibitory effect of compounds from flavone and isoflavone, we concluded that the glycosylation at the C8 position increased the inhibitory effect on M^pro^.

In the diarylheptanoid group, the order of inhibitory effects was as follows: bisdemethoxycurcumin < curcumin < dimethylcurcumin. In this group, curcumin contained two methoxy groups (C2′ and C4”) and showed higher inhibitory activity on M^pro^ than bisdemethoxycurcumin which lacked the methoxy group. However, its inhibitory activity was lower than that of dimethylcurcumin, which contained one methoxy group in C2′.

### 2.5. Combinative Inhibitory Effects on SARS-CoV-2 M^pro^ of Polyphenols

Herbal medicines have been used to treat COVID-19 in China [16]. According to the National Administration of Traditional Chinese Medicine, herbal medicines not only showed the ability to prevent COVID-19 infection in healthy persons but also improved the state of health in treated patients [16]. Most of the clinical studies used herbal extract mixtures [11]. Tannic acid found in red wines, herbaceous, legumes, sorghum, bananas, raspberries, and persimmons, belongs to the tannin family. The concentrations of tannins in red wines were estimated at 5~100 mM. Tannic acid inhibited not only M^pro^ SARS-CoV2 but also transmembrane protease serin 2 (TMPRSS2) [28]. Although tannic acid inhibits two different SARS-CoV2 enzymes, it has been reported to form complexes with protein, starch, and digestive enzymes causing the nutritional value to decrease [29,30,31]. Therefore, reducing the amount of tannic acid may reduce its side effects. In this study, we tested the effect of different combinations of tannic acid with myricetin, puerarin, and/or daidzein on M^pro^ activity. The concentration of tannic acid was fixed at 5 μM, while the concentrations of myricetin, puerarin, and daidzein were fixed at 20 μM. The inhibitory effects of mixing tannic acid with myricetin, puerarin, and/or daidzein on M^pro^ are shown in Figure 4. The inhibitory effects of a single polyphenol on M^pro^ were 30 ± 1% at 5 μM tannic acid, 23 ± 5% at 20 μM puerarin, 27 ± 3% at 20 μM daidzein, and 34 ± 1% at 20 μM myricetin. The inhibitory effects increased to 41 ± 2% at 5 μM tannic acid with 20 μM puerarin, 50% at 5 μM tannic acid with 20 μM daidzein, and 58 ± 2% at 5 μM tannic acid with 20 μM myricetin. The M^pro^ inhibitory activities of flavonoids at 45 μM were 50.8 ± 2% for puerarin, 43% for daidzein, and 50.5 ± 4% for myricetin, while they were 62 ± 6% at 5 μM tannic acid with 20 μM puerarin and 20 μM daidzein, 69 ± 2% at 5 μM tannic acid with 20 μM puerarin and 20 μM myricetin, and 69 ± 2% at 5 μM tannic acid with 20 μM puerarin and 20 μM myricetin. The highest inhibitory activity on M^pro^ (77 ± 1%) was achieved at 5 μM tannic acid with 20 μM puerarin, 20 μM daidzein, and 20 μM myricetin.

## 3. Discussion

The SARS-CoV-2 M^pro^ is one of the best-characterized targets for antiviral drug discovery. It plays an essential role in virus replication by digesting the viral polyproteins at more than 11 sites on the large polyprotein 1ab. The recognition sequence at most sites is LQ↓(S, A, G) (the cleavage site is indicated by ↓) [32] similar to that in SARS-CoV 3CL^pro^ [33,34]. Since inhibiting M^pro^ activity will block viral replication, we selected SARS-CoV2 M^pro^ as the target protein and Dabcyl-KTSAVLQ↓SGFRKME-Edans as its commercially available substrate. The biochemical characterization of M^pro^ showed that the optimum temperature, pH, and *K_m_* value were 37 °C, pH 7.5, and 16 μM, respectively. The *K_m_* value of the purified M^pro^ was similar to that of the SARS-CoV 3CL^pro^ expressed in *Pichia pastoris* with 15 ± 1 μM [20] and to that of the SARS-CoV 3CL^pro^ expressed in *E. coli* with 17 ± 4 μM [35]. We tested the influence of increasing concentrations of DMSO up to 50% (*v/v*). At up to 10% (*v/v*), we found no significant effects on SARS-CoV2 M^pro^ activity, but over 10% (*v/v*), DMSO significantly inhibited M^pro^ activity. At 50% (*v/v*) DMSO, M^pro^ lost its activity. No significant effect of 10% (*v/v*) DMSO on SARS-CoV2 M^pro^ activity was observed, similar to that on SARS-CoV 3CL^pro^ activity [34].

Polyphenols are a large family of phytochemicals with great chemical diversity as well as potential therapeutic diversity. Many polyphenols act as multi-target agents of high biochemical specificity and chemical diversity with lower cost, more covering mechanism, and minimal side effects. As one of the plant derivate polyphenols, curcumin down-regulated the coactivator of HBV transcription, peroxisome proliferator-activated receptor-gamma coactivator 1-α (PGC-1α), and inhibited HBV gene replication and expression [36]. In another example, tannic acid inhibits two SARS-CoV2 enzymes [28]. Thi et al. reported that epigallocatechin gallate or gallocatechin gallate containing galloyl moiety at 3-OH of the C-ring was required for inhibitory activity on SARS-CoV 3CL^pro^ [20]. Thus, polyphenols may be novel antiviral candidate compounds and lead-structures. In addition, polyphenols are inexpensive with minimal side effects. Polyphenols are known to be bioactive compounds of foods, species, medicinal plants, and nutraceuticals. Polyphenol intake varies within 50–1000 mg/day among different countries such as France (283–1000 mg/day), Spain (500–1100 mg/day), Italy (700 mg/day), Finland (890 mg/day), Brazil (534 mg/day), Japan (1500 mg/day), US (240–350 mg/day), China (50–500 mg/day), and Korea (320 mg/day) [37,38,39]. Thus, polyphenols have potential use in the prevention of SARS-CoV2. Numerous studies on herbal medicines against SARS-CoV-2 have been conducted in vitro, in vivo, and in ovo [16,40,41]. However, most clinical studies were performed with food or herb combinations based on the traditional Chinese formulas. Lianhuaqingwen is a Chinese patent medicine composed of 13 herbs [40,42]. It inhibited SARS-CoV-2 replication in a dose-dependent manner with an IC_50_ of 411 μg/mL, reduced the production of pro-inflammatory cytokines, and affected particle morphology of the virus [40]; 61 polyphenols were identified in this herbal mixture. Therefore, we aimed to study the effects of bioactive compounds in herbal extracts on SARS-CoV-2 M^pro^ as well as their structure–activity relationship. Garlic is a popular medicinal food owing to its numerous health benefits (anticancer, antibacterial, antiviral, antidiabetic, antihypertensive, antioxidative, and immunity-enhancing benefits) [43]. However, raw garlic causes gastrointestinal discomfort and has a pungent taste and smell, resulting in a decline in its consumption. Black garlic, with a sweet and sour taste [43], overcame this drawback of raw garlic. Black garlic has more reducing sugar (80-fold), lipids (3.2-fold), organic acids (3.9-fold), total phenolic acids (~7.8-fold), flavonoids (~3.5-fold), polyphenols (1.8-fold), alkaloids (28.6-fold), and vitamins (1.3-fold) than raw garlic [17,44]. Also, chlorogenic acid, vanillic acid, and quercetin were newly synthesized, whereas the amount of other compounds, such as gallic acid, caffeic acid, coumaric acid, ferulic acid, catechin, epicatechin, epigallocatechin gallate, myricetin, resveratrol, and morin increased from 1.1- to 26.2-fold, which contributed to the enhancing of total phenolic acid and total flavonoid content by up to 7.8- and 3.5-fold, respectively [17]. From these results, we found that the inhibitory effect of black garlic on M^pro^ was possibly owing to its phenolic acids (gallic, chlorogenic, caffeic, and ferulic) and flavonoid compounds (myricetin, quercetin, epigallocatechin gallate, epicatechin, catechin, and resveratrol) [17].

The COVID-19 pandemic started in December 2019 and has extended until now; there are no approved therapeutics for this disease. Although numerous researchers performed computational chemistry [45] and virtual screening of FDA-approved drugs [46] and flavonoid compounds [47,48,49,50] with M^pro^, their efficiencies against M^pro^ need to be confirmed by in vitro and in vivo experiments. To the best of our knowledge, among our tested polyphenols, EGCG and tannic acid have been reported to have inhibitory effects on M^pro^ with IC_50_ values of 7.6 μg/mL and 2.1 μM, respectively [23,45]. Here, we found that the EGCG and tannic acid inhibited M^pro^ with IC_50_ values of 171 μM and 9 μM, respectively (Table 1). The differences in the results may be due to different substrate and reaction conditions used in each study. For example, Jang et al., (2020) performed the EGCG inhibition assay at 37 °C for 5 h using the same substrate; however, Coelho et al., (2020) conducted the inhibition assay using MCA-AVLQSGFR-K(Dnp)-K-NH_2_. Although we did not perform experiments to study the binding mode of these compounds by molecular docking, there are several reports that predict the inhibition mode of these compounds [48,49,50]. Molecular docking studies with M^pro^ protein showed that some compounds, such as rutin (–9.2 kcal/mol), hesperidin (–8.1 kcal/mol), naringin (–8.1 kcal/mol), EGCG (–7.9 kcal/mol), catechin (–5.8 kcal/mol), myricetin (–5.9 kcal/mol), and caffeic acid (–5.1 kcal/mol), had the highest binding energy and hydrogen-bonding interactions with key amino acid residues of M^pro^ [31,48,50]. Our experiments confirmed that these compounds inhibited M^pro^ (Table 1). Although the polyphenols assessed in this study showed moderate inhibition against M^pro^, they may prevent or at least impede SARS-CoV-2 infection. COVID-19 therapies can be divided into two types depending on their targets [51], one is designed to boost the human immune system or inhibit the inflammatory process, and the other to directly attack the virus by hindering its replication and entry into the human cell. Polyphenols have been reported to affect the level and composition of immunoglobulins, inflammation, and immune cell populations [52]. For example, polyphenols such as curcumin, epigallocatechin gallate, naringenin, kaempferol, apigenin, and resveratrol decrease pro-inflammatory cytokines, according to in vitro and in vivo studies [46]. Thus, polyphenols not only inhibit SARS-CoV2 M^pro^ activity but also affect the human immune system. In addition, polyphenols are the most common and the largest plant compounds in the human diet. They are extracted from plants and found in foods and beverages. The dietary intake of polyphenols has been estimated to vary from 100 to 1000 mg/day. Further studies of foods containing these polyphenols against SARS-CoV-2 are needed.

## 4. Materials and Methods

### 4.1. Preparation of Active SARS-CoV-2 M^pro^

SARS-CoV-2 M^pro^ gene was codon optimized and synthesized for expression in *Escherichia coli* based on M^pro^ amino acid sequence (GenBank accession MT483553.1: 3264–3569 aa) by Genscript (Piscataway, NJ, USA). M^pro^ was inserted into the pET28a vector (pET28a-M^pro^) (Novagen, Darmstadt, Germany) for overexpression of M^pro^ protein with polyhistidine tags. pET28a-M^pro^ was transformed into *E. coli* BL21(DE3) using the heat shock method and then spread on LB agar containing (50 μg/mL) (Sigma). The plate was inoculated at 37 °C overnight. LB supplemented with kanamycin was used for a single colony grown at 37 °C. One mM isopropyl β-d-1-thiogalactopyranoside (IPTG) (Sigma) was used to induce M^pro^ when the optical density (OD_600_) became 0.5. Induced cells were cultured at 16 °C with shaking (200 rpm) for 12 h. Cells were harvested by centrifugation (8000× *g* for 30 min at 4 °C), resuspended in 50 mM Tris-HCl pH 7.0, and lysed by sonication (Ultrasonic processor 250, Sonics and Materials Inc., Newtown, CT, USA). After sonication, cell lysate was obtained by centrifugation (12,000× *g* for 30 min) and then loaded onto 12 mL of Ni-Agarose resin (Qiagen, Hilden, Germany). A buffer containing 50 mM Tris-HCl, 100 mM NaCl, 300 mM imidazole, pH 8.0 was used to elute proteins from the Ni-Agarose column. Fractions containing purified proteins were obtained and dialyzed against a 50 mM Tris-HCl buffer (pH 7.5). Protein concentration was determined using a protein determination kit (Bio-Rad protein assay kit, Bio-Rad Laboratories, Hercules, CA, USA), and 12% SDS-PAGE was used to confirm the purity of the protein.

Proteolytic activity of M^pro^ was measured by using fluorogenic peptide (Dabcyl-KTSAVLQ↓SGFRKME-Edans) (FRET) as a substrate (Nguyen et al., 2012). The reaction mixtures composed of 20 μM of FRET substrate and 8 μg of M^pro^ in 50 mM Tris-HCl (pH 7.5) were run at 37 °C for 30 min. Relative fluorescence units (RFUs) were verified with SpectraMax M3 (Molecular Devices, San Jose, CA, USA) at excitation and fluorescence emission wavelengths of 355 and 538 nm, respectively.

### 4.2. Biochemical Characteristics of the Active SARS-CoV-2 M^pro^

#### 4.2.1. Effect of pH on Activity and Stability of SARS-CoV-2 M^pro^

The effect of pH on M^pro^ activity was studied at different pHs ranging from pH 3.0 to 11.0 using the following buffers: glycine-HCl (50 mM, pH 3.0), sodium acetate buffer (50 mM, pH 5.0–6.5), Tris-HCl buffer (50 mM, pH 7.0–9.0), and glycine-NaOH buffer (50 mM, pH 9.5–11.0) [53]. The optimum pH was determined by incubation of M^pro^ in each buffer for 30 min using 20 µM of FRET as a substrate at 37 °C for 30 min. RFUs were recorded as described above.

The pH stability of M^pro^ was carried out by incubation of M^pro^ for 24 h in the chosen buffers at 4 °C. Then, the pH of M^pro^ was accustomed to 7.5 by membrane dialysis. M^pro^ activity was carried out with 20 µM of FRET at 37 °C for 30 min. RFUs were recorded as described above.

#### 4.2.2. Effect of Temperature on Activity and Stability of SARS-CoV-2 M^pro^

The effect of temperature on M^pro^ activity was performed by reacting a reaction mixture composed of M^pro^ enzyme and 20 µM of FRET in 50 mM Tris-HCl (pH 7.5) at various temperatures from 4 to 70 °C. RFUs were recorded as described above.

M^pro^ temperature stability was performed by keeping the enzyme from 4 to 60 °C for 24 h. Then, the enzyme was added to the reaction digest containing 20 µM of FRET in 50 mM Tris-HCl (pH 7.5). The reactions were done at 37 °C for 30 min. RFUs were verified with SpectraMax M3 as described above.

#### 4.2.3. Effect of DMSO on SARS-CoV-2 M^pro^ Activity

The effect of dimethyl sulfoxide (DMOS) on M^pro^ activity was measured by adding different concentrations of DMSO from 0 to 50% (*v/v*) to the reaction mixture comprising M^pro^ in 50 mM Tris-HCl pH 7.5. The FRET substrate was added to initiate the reaction. The control was a reaction without adding DMSO. The reactions were conducted at 37 °C for 30 min. RFUs were monitored as described above.

#### 4.2.4. Kinetic Characterization

The kinetic parameter of M^pro^ was added to different concentrations of FREP substrate (2.5–45 μM) to the reaction mixture composed of M^pro^ in 50 mM Tris-HCl pH 7.5. The reactions were run at 37 °C for 8 min. Reaction results were linear within this time. The Michaelis–Menten constant (*K_m_*) was obtained from the Lineweaver–Burk plot using the Sigmaplot program (Systat Software, San Jose, CA, USA).

### 4.3. Inhibition Assay

Black garlic extract and 49 compounds that were selected for M^pro^ inhibitory activity are listed in Table 1. A 16 μM FRET substrate was added to the reaction digest containing M^pro^ and a 200 μM inhibitor or 0.5 mg/mL of black garlic extract in 50 mM Tris-HCl (pH 7.5) to start a reaction. The reaction was run at 37 °C for 15 min and RFUs were recorded as above. The inhibition activity of M^pro^ was calculated as follows:Inhibition activity (%) = 100 − [(S − S_o_) / (C − C_o_)] × 100(1)
where C_o_ and C were the RFUs of the control (buffer, enzyme, and substrate) after 0 and 15 min of reaction and S_o_ and S were the RFUs of the assay sample (buffer, inhibitor, enzyme, and substrate) after 0 and 15 min of reaction. Fifteen compounds and black garlic extract were selected for the 50% inhibitory concentration (IC_50_) determination. IC_50_ was the M^pro^ inhibitor concentration necessary to reduce 50% M^pro^ activity compared to the reaction without adding an inhibitor. To determine IC50, 16 μL FRET substrate was added to the reaction digest containing M^pro^ and different concentrations of inhibitors (1–400 μM) in 50 mM Tris-HCl (pH 7.5) to start the reaction as described above.

### 4.4. Combinative Inhibitory Effects of Selected Compounds

The combinative inhibitory effects of tannic acid, puerarin, daidzein, and myricetin were performed by adding 5 μM of tannic acid with 20 μM of puerarin, and/or daidzein, and/or myricetin to a reaction mixture containing a 16 μM FRET substrate and M^pro^ in 50 mM Tris-HCl (pH 7.5) as described above.

### 4.5. Statistical Analysis

All experiments were performed in triplicate, and results were expressed as the mean ± standard deviation. Statistical comparisons were made by one-way analysis of variance followed by Tukey’s comparison tests on GraphPad Prism 8 (San Diego, CA, USA). Values were considered to be significant when *p* < 0.05.

## 5. Conclusions

Here, we reported the overexpression and biochemical characterization of an active SARS-CoV-2 M^pro^. The purified M^pro^ was used to study the inhibitory effect of black garlic extract and 49 polyphenols. Black garlic extract and 15 polyphenols showed the inhibition of M^pro^ activity with IC_50_ values of 137 μg/mL and from 9 to 197 μM, respectively. From the structure–activity relationship of these polyphenols, we found that the hydroxyl group in C3′, C4′, C5′ in the B-ring, C3 in the C-ring, C7 in the A-ring, the double bond between C2 and C3 in the C-ring, and glycosylation at C8 in the A-ring contributed to the inhibitory activity of flavonoids on M^pro^. Different combinations of tannic acid with puerarin, daidzein, and/or myricetin also increased inhibitory effects on M^pro^. Polyphenols are products extracted from plants and found in foods and beverages such as fruits, vegetables, tea, wine. Therefore, further studies of foods containing these polyphenols are needed.

## Figures and Tables

**Figure 1 molecules-26-01924-f001:**
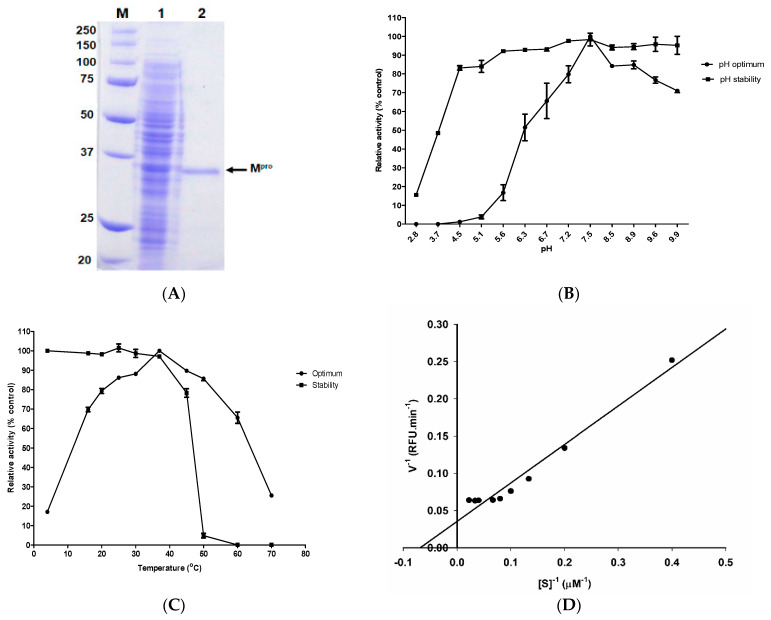
SDS-PAGE analysis (**A**), optimum pH and pH stability (**B**), optimum temperature and temperature stability (**C**), and Lineweaver–Burk plot for determination of *K_m_* value of purified SARS-CoV-2 M^pro^ (**D**). (**A**) Lane M: the molecular mass markers, lane 1: cell lysate with 1 mM IPTG induction to overexpress M^pro^, lane 2: purified M^pro^ after using the Ni-NTA column chromatography. (**B**) Buffers used: glycine-HCl (pH 3.0), sodium acetate buffer (pH 5.0–6.5), Tris buffer (pH 7.0–9.0), and glycine-NaOH buffer (pH 9.5–11.0). For pH stability, the enzyme was kept at 4 °C for 24 h under various pH conditions (pH 3.0–11.0). (**C**) For thermal stability, the enzyme was kept from 4 to 60 °C for 24 h. The enzyme activity was measured with 20 µM of FRET as a substrate at 37 °C for 30 min. (**D**) The kinetic parameter of M^pro^ was added to different concentrations of FREP substrate (2.5–45 μM) to the reaction mixture composed of M^pro^ in 50 mM Tris-HCl pH 7.5. The reactions were run at 37 °C for 8 min.

**Figure 2 molecules-26-01924-f002:**
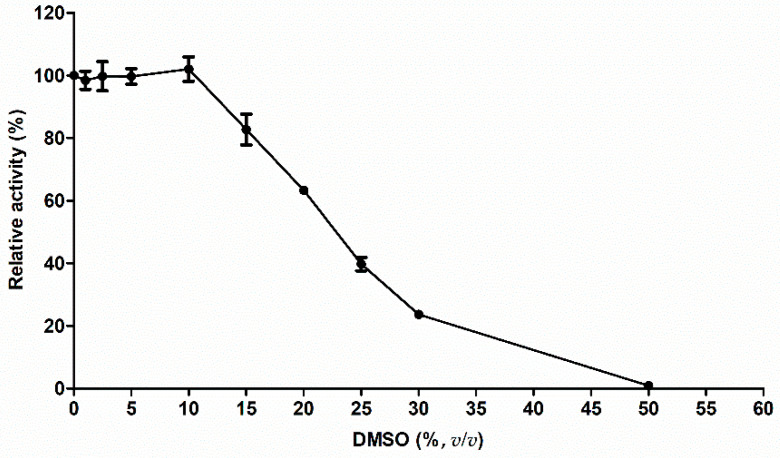
Effect of DMSO on SARS-CoV-2 M^pro^ activity. M^pro^ was incubated for 30 min at 37 °C in 50 mM Tris-HCl buffer pH 7.5 containing 20 µM of FRET with different concentrations of DMSO (0–50%, *v/v*).

**Figure 3 molecules-26-01924-f003:**
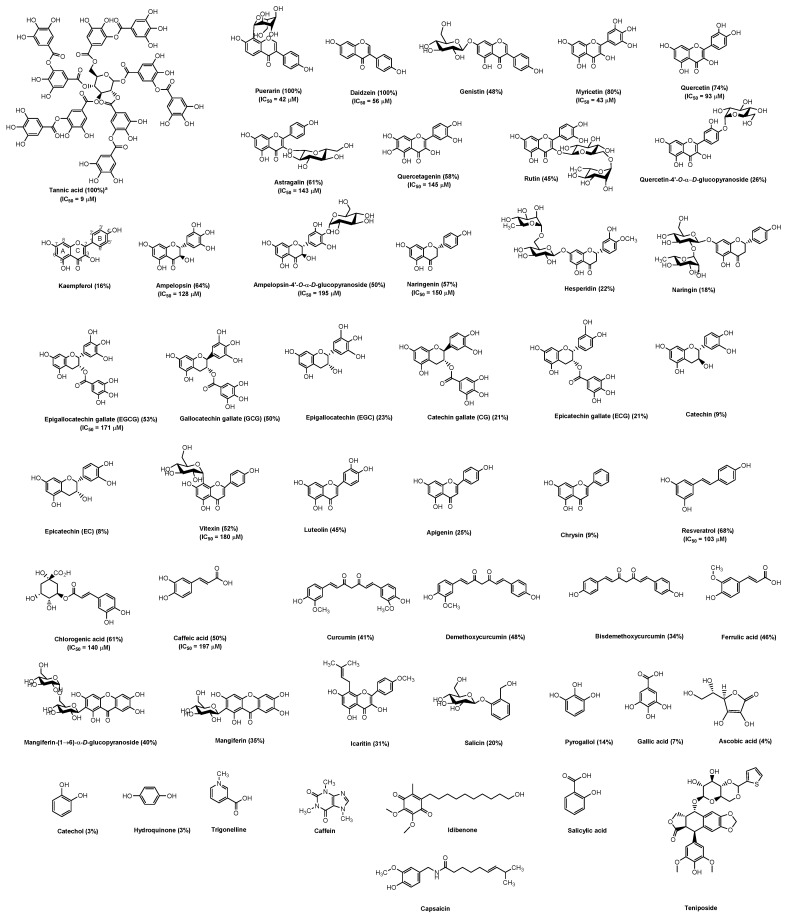
Chemical structure of plant derivate polyphenols against SARS-CoV-2 M^pro^.

**Figure 4 molecules-26-01924-f004:**
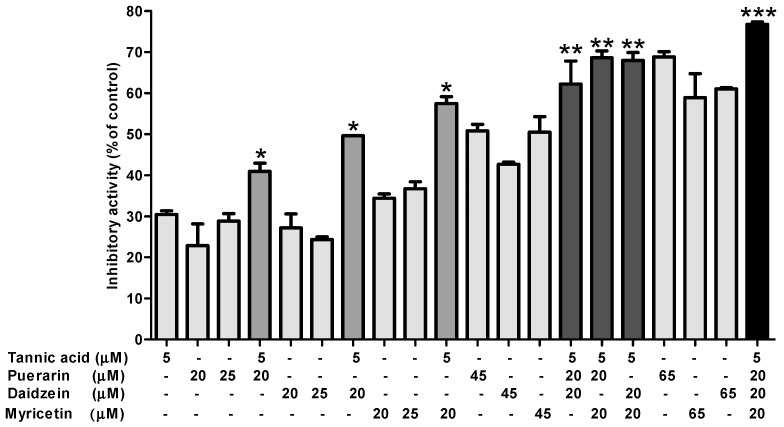
Inhibitory effects on SARS-CoV-2 M^pro^ of mixtures of tannic acid with puerarin, daidzein, and/or myricetin. All tests were performed in triplicate, and bars indicate the mean and standard deviation. *, **, ***: *p* < 0.05 compared to 25 μM, 45 μM, 65 μM of each puerarin, daidzein, and myricetin, respectively.

**Table 1 molecules-26-01924-t001:** The inhibitory effect of polyphenol compounds on SARS-CoV-2 M^pro.^

No	Compound	Group	Source	Inhibition (%)	IC_50_ (μM)
1	Black garlic extract	Mixtures	SerimFood	100	137 ± 10 μg/mL
2	Tannic acid	Tannoid	Sigma	100	9
3	Puerarin	Isoflavone	Sigma	100	42 ± 2
4	Daidzein	Isoflavone	Sigma	100	56
5	Genistin	Isoflavone	Sigma	48	ND
6	Myricetin	Flavonol	Sigma	80	43 ± 1
7	Quercetin	Flavonol	Sigma	74	93 ± 5
8	Astragalin	Flavonol	Amore Pacific	61	143 ± 9
9	Quercetagenin	Flavonol	Sigma	58	145 ± 6
10	Rutin	Flavonol	Sigma	45	ND
11	Quercetin-4′-*O*-α-d-glucopyranoside	Flavonol	Synthesized	26	ND
12	Kaempferol	Flavonol	Sigma	16	ND
13	Ampelopsin	Flavanonol	Sigma	64	128 ± 5
14	Ampelopsin-4′-*O*-α-d-glucopyranoside	Flavanonol	Synthesized	50	195 ± 5
15	Naringenin	Flavanone	Sigma	57	150 ± 10
16	Hesperidin	Flavanone	Sigma	22	ND
17	Naringin	Flavanone	Sigma	18	ND
18	Epigallocatechin gallate (EGCG)	Flavan-3-ols	Sigma	53	171 ± 5
19	Gallocatechin gallate	Flavan-3-ols	Sigma	50	ND
20	Epigallocatechin (EGC)	Flavan-3-ols	Sigma	23	ND
21	Catechin gallate (CG)	Flavan-3-ols	Sigma	21	ND
22	Epicatechin gallate (ECG)	Flavan-3-ols	Sigma	21	ND
23	Catechin	Flavan-3-ols	Sigma	9	ND
24	Epicatechin (EC)	Flavan-3-ols	Sigma	8	ND
25	Vitexin	Flavone	Sigma	52	180 ± 6
26	Luteolin	Flavone	Sigma	45	ND
27	Apigenin	Flavone	Sigma	25	ND
28	Chrysin	Flavone	Sigma	9	ND
29	Resveratrol	Stilbenoid	Sigma	68	103 ± 6
30	Chlorogenic acid	Hydrocinnamic acid	Sigma	61	140
31	Caffeic acid	Dihydroxycinnamic acid	Sigma	50	197 ± 1
32	Demethoxycurcumin (DMC)	Diarylheptanoid	TCI chemical	48	ND
33	Bisdemethoxycurcumin (BDMC)	Diarylheptanoid	TCI chemical	34	ND
34	Curcumin	Diarylheptanoid	TCI chemical	41	ND
35	Ferulic acid	Hydrocinnamic acid	Sigma	46	ND
36	Mangiferin-(1- > 6)-α-d-glucopyranoside	Xanthonoid	Synthesized	40	ND
37	Mangiferin	Xanthonoid	Sigma	35	ND
38	Icaritin	Favonol	Sigma	31	ND
39	Salicin	Salicylate	Sigma	20	ND
40	Pyrogallol	Benzenetriol	Sigma	14	ND
41	Gallic acid		Sigma	7	ND
42	Ascorbic acid		Sigma	4	ND
43	Catechol	Bezenediol	Sigma	3	ND
44	Hydroquinone		Sigma	3	ND
45	Trigonelline	Alkaloid	Sigma	1	ND
46	Caffein	Alkaloid	Sigma	0	ND
47	Idebenone	1,4-benzoquinone	TCI chemical	0	ND
48	Salicylic acid	Monohydroxybenzoic acid	Sigma	0	ND
49	Capsaicin	Capsaicinoid	Sigma	0	ND
50	Teniposide	Podophyllotoxin	Sigma	0	ND

## Data Availability

All data are within this manuscript.

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
