# Peer review of "The Inhibitory Effects of Plant Derivate Polyphenols on the Main Protease of SARS Coronavirus 2 and Their Structure–Activity Relationship"

_molecules, 2021, doi:10.3390/molecules26071924_

Round 1
Reviewer 1 Report
This paper has several problems. One is English language usage. This is often problematic in this paper, and, at times, becomes a problem for readers to understand.
Here are some examples of usage problems.
page 2 line 52
At present, there is no antiviral drugs on market ..
should be
At present, there are no antiviral drugs on the market ..
Page 2 line 56-7
Nonetheless, their efficacy against COVID-19 infection leftovers unclear.
What does the word leftovers mean in this context?
page 2 line 70, 71
In our previous study, polyphenols such as ampelopsin, puerarin, quercetin, daidzein, epigallocatechin gallate (EGCG), and gallocatechin gallate (GCG) were inhibited against 3CLpro of SARS-CoV [20]
What do the words were inhibited against mean? Does it mean that these compounds inhibited 3CLpro of SARS-CoV?
P. 9 line 181
Change this:
... medicines were not only showed prevention of COVID-19 infection in healthy persons but also improved health state of treated patients ..
to this:
...medicines not only showed the ability to prevent COVID-19 infection in healthy persons but also improved the state of health in treated patients.
Page 4, 221
It is a Chinese patent medicine composed of 13 herbs [34]
What is the name of this Chinese patent medicine?
In Table I, tannic acid is misspelled.
Here are comments on presentation and on the content itself:
1. It would be helpful if the compounds in Table I were ordered by IC50
2. Units are missing in the graph in the supplementary materials. It would appear from the graph that the substrate was used at a level over 1 M concentration. This cannot be correct.
3. on p. 14 lines 355-6 it says that:
The mixture of tannic acid with puerarin, daidzein, and/or myricetin increased inhibitory effects against Mpro.
Did the calculation of Ki reveal any effects of synergy due to the mixture, or only the increased inhibition due to the mixture itself? That is, was the mixture better than its component parts?
4. It would be helpful to attach the IC50 values to the compound names in Figure 3.
5. This reviewer notes that the IC50 values do not show very strong binding. 8 uM is the best, when most IC50 values range from 40-50 micromolar or higher. These values are those that would be expected for PAINS compounds.
The authors should discuss which of their compounds are generally accepted PAINS compounds. Some of their compounds are given in an accepted list of PAINS:
See Table II in https://europepmc.org/article/PMC/4791574
If any compounds used here are listed by others as PAINS compounds, then that fact must be discussed in this paper.
6. Are the concentrations of the compounds under study, when present in foodstuffs, sufficient to generate antiviral effects? That is alluded to, but not calculated in a convincing way that includes estimates of body mass and volume.
7. The authors present no evidence to support the last sentence in the abstract and the penultimate sentence on line 268.
Therefore, it would be used to prevent COVID-19 and improvement for human health.
These sentences and any others like it must be deleted.
Reviewer 2 Report
The peer-reviewed article presents a study of the action of black garlic polyphenols on the major protease (Mpro) involved in SARS-CoV-2 viral replication. This kind of research is important, since phytopreparations can not only provide prevention of COVID-19 infection in healthy people, but also improve the course of patient's recovery. The strongest aspect of the study is the study of the structure-activity relationship for Mpro inhibitory activity.
I think that the article can be improved taking into account the following comments and suggestions:
- Some values are given in the article in the form of 4 significant digits. For example, IC50 is 142.8±9.4 mM for Astragalin (Table 1). It is more realistic to give the value of 143±10. It is clear that it is impossible to determine such values with the greater accuracy, so the presented in the manuscript values should be rounded up to 3 digits.
- It is better to represent the dependence shown in Figure 1D in Lineweaver-Burk coordinates. It will then be seen whether the data are represented as a straight line. And it is imperative to give the error interval for the Km.
- Authors write that the Mpro was purified that was confirmed on SDS-PAGE. Is it possible to give a numerical value characterizing the degree of purity?
- Authors write that Mpro was stable in the presence of 0-10% DMSO. Is it possible in Discussion to cite literary data on the stability of other proteases (enzymes) in this water-organic system? Is the stability of Mpro high in comparison with other proteases?
- Lines 214,215. Please, provide the value of Km for 3CLpro from link [20].
- Lines 220,221: “Runfeng et al. (2020)”. Please, give the link in number.
- Line 246: “171.49”. This number is missing from Table 1.
